# Extending Deep Learning Emulation Across Parameter Regimes to Assess Stochastically Driven Spontaneous Transition Events

**Ira J. S. Shokar**[*]**, Peter H. Haynes, Rich R. Kerswell**
Department of Applied Mathematics and Theoretical Physics, University of Cambridge,
Wilberforce Road, Cambridge, CB3 0WA, UK
`*i.j.s.shokar@damtp.cam.ac.uk`

## Abstract

Given the computational expense associated with simultaneous multi-task learning, we leverage fine-tuning to generalise a transformer-based neural network emulating a stochastic dynamical system across a range of parameters. Fine-tuning a neural network with a dataset containing a set of parameter values yields a 40-fold reduction in required training size compared to training ab initio training for each new parameter. This facilitates rapid adaptation of the deep learning model, which can be used subsequently across a large range of the parameter space or tailored to a specific regime of study. We demonstrate the model's ability to capture the relevant behaviour even at interpolated parameter values not seen during training. Applied to a well-researched zonal jet system, the speed-up provided by the deep learning model over numerical integration and the ability to sample from the probabilistic model makes uncertainty quantification in the form of statistical study of rare events in the physical system computationally feasible. Our code is available at `https://github.com/Ira-Shokar/Stochastic-Transformer`.

## 1 Introduction

Given the computational complexities inherent in turbulence modelling, investigations have delved into cost-effective alternatives to physics-based solvers. Machine learning has emerged as a promising direction, given its capacity to learn non-linear transformations and delegate a substantial portion of the requisite computation to offline training (Vinuesa & Brunton, 2022).

Most machine learning methods, however, face challenges in accurately generalising beyond their training distribution. An example would include emulation of a dynamical system, in which physical parameters governing the PDE (Partial differential equation) differ from those seen during training. One approach to overcome this is transfer learning (Pan & Yang, 2010). In the case of PDEs, learned weights of a base trained network are frozen and additional layers are trained to generalise to the new tasks (Pellegrin et al., 2022; Goswami et al., 2022; Chen et al., 2021). Alternatively, the transformer architecture (Vaswani et al., 2017) has been shown to perform extremely well in generalisation tasks without additional layers. This is due to the ability of the attention mechanism to conduct self-supervised learning during pre-training (Devlin et al., 2019).

In this work, we pre-train a transformer based network on a dataset comprising trajectories generated from a single parameter of a PDE, to learn a representative basis that captures the underlying dynamics of a stochastic dynamical system. This is followed by fine-tuning on a smaller dataset comprising trajectories from a range of parameter values, to generalise across parameter regimes. The neural network provides a resource-efficient emulator of the PDE. We show that autoregressive roll-outs remain indefinitely stable, while the neural network is capable of capturing the statistical properties across parameter regimes.

## 2 Methodology

### 2.1 Physical System - Quasi-Geostrophic Turbulence

In the domain of earth system modeling, parameterisation schemes often integrate stochastic elements to approximate the impact of small-scale processes not explicitly resolved by the model, thus capturing their inherently chaotic nature (Berner et al., 2015). In this study, we introduce a deep learning emulator spanning a parameter range of a well-studied idealised Geophysical Fluid Dynamics (GFD) system—beta-plane turbulence (Rhines, 1975; Galperin & Read, 2019). This system serves as a useful simplest model for many aspects of flows in atmospheres and oceans. Modelled as a single-layer flow on a doubly-periodic domain, turbulence is generated via stochastic forcing, $\xi(x, y, t)$, in the vorticity equation:

$$\partial_t \zeta + u \cdot \nabla \zeta + \beta \partial_x \psi = \xi - \mu \zeta + \nu_n \nabla^{2n} \zeta \in \mathbb{R}^{N_x \times N_y} \tag{1}$$

with velocity field $u(x, y, t) = (-\partial_y \psi, \partial_x \psi)$ (note the sign convention used widely in GFD), stream function, $\psi$ and the relative vorticity $\zeta = \partial_x v - \partial_y u$. The parameter $\beta$ represents the Rossby parameter, the rate of planetary rotation at a given latitude. Energy dissipation occurs through linear damping at a rate $\mu$, while hyperviscosity, characterised by a coefficient $\nu$ and order $n$, removes vorticity at small scales. The forcing term $\xi(x, y, t)$ takes the form of white-in-time noise injected into an annulus in wavenumber space. Simulations are run at a resolution $N_x = N_y = 256$ - see Appendix A.1 for further details.

In turbulence modelling, a Reynolds decomposition is typically employed as an averaging procedure to separate coherent structures from fluctuations, here that takes the form of an eddy-mean decomposition in the zonal direction to separate the dynamics: $u(x, y, t) = \overline{u}(y, t) + u'(x, y, t)$ (where $\overline{u}(y, t) = \frac{1}{L_x} \int_0^{L_x} u \, dx$ and $L_x = 2\pi$ is the domain size in $x$). This gives equations for the zonally averaged zonal velocity field $U(y, t) = \overline{u}(y, t)$ and the associated eddy fluctuation fields, $\zeta'(x, y, t)$:

$$\partial_t U = \left(-\mu + \nu_n \partial_y^{2n}\right) U + \overline{\zeta' v'} \in \mathbb{R}^{N_y} \tag{2}$$

$$\partial_t \zeta' = \left(-\mu + \nu_n \nabla^{2n} - U \partial_x\right) \zeta' + (\partial_{yy} U - \beta) v' + \xi + EENL \in \mathbb{R}^{N_x \times N_y} \tag{3}$$

where $EENL = \partial_y(\overline{\zeta' v'}) - \partial_y(\zeta' v') - \partial_x(\zeta' u')$ denote the eddy-eddy nonlinear interaction terms. Here we see that the evolution of $U$ depends on the nonlinear two-way interaction between the eddies and the mean flow, given by the Reynolds stress term ($\overline{\zeta' v'}$). Forcing noise, $\xi$, is solely applied to the eddy fields, and therefore its effect on the mean flow is indirectly filtered through non-linear processes. The influence of $\beta$, akin to $\xi$, also manifests indirectly through the Reynolds stress term.

As with most PDEs, varying the parameters of the governing equation gives rise to diverse phenomena. In this system, varying the parameter $\beta$ gives rise to a range of range of dynamics, seen in the latitude-time plots in Figure 1, in which not just the number of jets varies (maxima in $U$ - yellow in colour), but also the dynamic tendencies, namely frequency of nucleation of a new jet, coalescence of two jets, and latitudinal jet translation.

### 2.2 Stochastic Transformer

We train a deep learning emulator of equation 2 tailored to accommodate varying values of $\beta$. We employ the Stochastic Transformer (ST) (Shokar et al., 2023), a transformer-based probabilistic model, that projects forward to predict an ensemble of plausible subsequent states $\{\tilde{U}_{t+1}\}$.

Physical models are generally formulated as PDEs that are first-order in time, however given we are not operating with full physical states, but with the coarse grained variable $U$, in line with other work (Price et al., 2023), we found it advantageous to condition on additional temporal context. The non-linear nature of equation 3 means that there is no closed form solution for $\zeta'$ and as such conditioning on additional temporal histories of $U$ can be thought of as providing an approximate a closure for Reynolds stress term in equation 2. Consequently, the ST is structured to take time histories of $U_{t-N:t}$, of length $N$, $\beta$ and Gaussian noise $\epsilon \sim \mathcal{N}(0, 1)$ as inputs - in practice we found only conditioning on the last $N = 2$ sequence members was required.

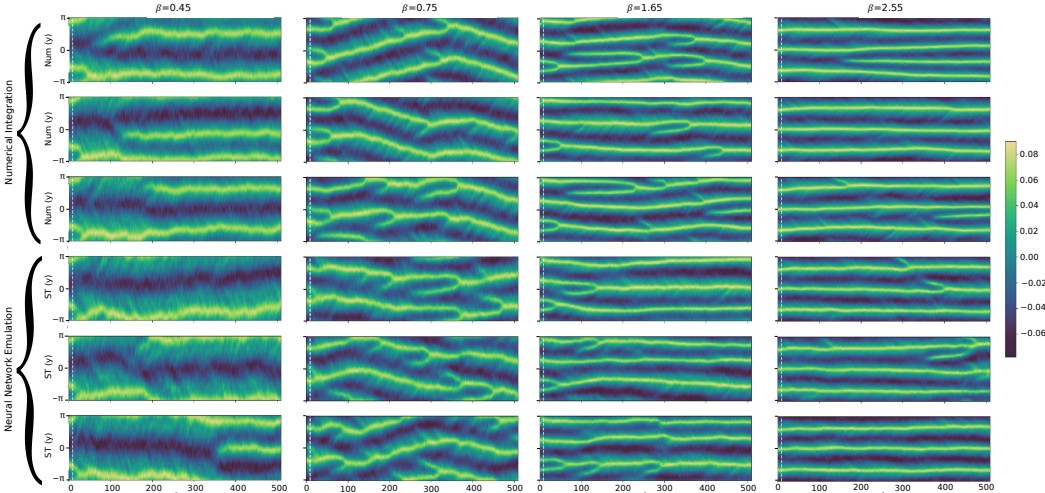

Figure 1: Latitude-time plots of the zonally-averaged flow $U(y,t)$ displaying ensemble of a numerical integration and neural network emulations across previously unseen parameter values. The ST is initially pre-trained for $\beta = 0.9$ using 200,000 snapshots, followed by fine-tuning for $\beta = \{0.3, 0.6, 1.2, 1.5, 1.8, 2.1, 2.4, 2.7\}$ with 5,000 snapshots for each value. The ST produces an ensemble of trajectories originating from identical initial conditions, showcased here over a duration of 500 time units, for the previously unseen parameter values of $\beta = \{0.45, 0.75, 1.65, 2.55\}$. The ST produces qualitatively plausible trajectories, adeptly generalising to the different regimes.

Generative models are often conditioned by prompt or by class information in which an encoder network is employed to provide an embedded representation (Ramesh et al., 2021). However, as the additional conditioning in physical systems comes in the form of continuous variables (here $\epsilon$ and $\beta$), we can include them directly via cross-attention. The ST outlined in Shokar et al. (2023) implements cross-attention to learn an adaptive state-dependent weighted contribution for stochastic noise $\epsilon$, given time histories $U_{t-N:t}$, to model the conditional probability $p(U_{t+1}|U_{t-N:t}, \epsilon)$. Sampling $\epsilon$ facilitates ensemble generation with different noise realisations, akin to sampling $\xi$ in equation 1.

In this work, we require the transformer to also account for the influence of the parameter, $\beta$, in modelling $p(U_{t+1}|U_{t-N:t}, \epsilon; \beta)$. To achieve this, we introduce $\beta$ as an additional feature preceding the $W^K \in \mathbb{R}^{(D+1)\times D}$ (key) and $W^V \in \mathbb{R}^{(D+1)\times D}$ (value) transformations, with $D = N_y - 256$. This differs from handling of $\epsilon$ which undergoes separate transformations via $W^{K\epsilon} \in \mathbb{R}^{D\times D}$ and $W^{V\epsilon} \in \mathbb{R}^{D\times D}$ before concatenation to facilitate a state-dependent adaptive weighing of the noise during attention. By incorporating $\beta$ directly into the linear transformations, we parameterise the model, independent of the current state $U_{t-N:t}$. Additionally, including temporal histories, $U_{t-N:t}$, within the key and value vectors to facilitate learning of temporal correlations, results the following:

$$Q = W^Q U_{t-N:t} \in \mathbb{R}^D \tag{4}$$

$$K = \{W^K \{U_{t-N:t}, \beta\}_d, W^{K\epsilon}\epsilon\}_s \in \mathbb{R}^D \tag{5}$$

$$V = \{W^V \{U_{t-N:t}, \beta\}_d, W^{V\epsilon}\epsilon\}_s \in \mathbb{R}^D \tag{6}$$

where each $W$ denotes a learnable weight matrices, $Q$ denotes the query vector and $\{.\}_s$, $\{.\}_d$ here represent concatenations over the sequence dimension and the feature dimensions respectively. Attention is calculated as standard: $\text{Attention}(Q, K, V) = A(Q, K)V = \text{softmax}\left(QK^T D^{-\frac{1}{2}}\right)V$, where softmax denotes the softmax operation to introduce sparsity in the attention matrix $A$.

Following the approach outlined in Shokar et al. (2023), we introduce noise $\epsilon$ solely in the initial of M transformer blocks, to only induce a single forcing when making a forecast of $U_{t+1}$, aligning with the way stochastic PDEs are typically forced. Consistent with conventional practice in employing cross-attention, all subsequent attention blocks still include conditioning on $\beta$, giving

$K = W^K \{U_{t-N:t}, \beta\}_d$, $V = W^V \{U_{t-N:t}, \beta\}_d$ and $Q = W^Q U_{t-N:t}$ in subsequent transformer blocks - see Appendices B.1 and B.2 for further details on the architecture and loss function.

## 2.3 Pre-training and Fine-tuning

The ST is initially pre-trained with a dataset of 200,000 snapshots where $\beta = 0.9$, followed by fine-tuning with $\beta = \{0.3, 0.6, 1.2, 1.5, 1.8, 2.1, 2.4, 2.7\}$ with 5,000 snapshots for each value. This range was chosen as above it dynamics display zonostrophy (persistent jets with minimal nucleation, coalescence, or jet translation), while below it, friction dominates, suppressing jet formation. The pre-trained weights are then used for initialisation with fine-tuning refining the model to generalise to the dynamics corresponding to various values of $\beta$. The initial learning rate used for the fine-tuning is a factor of 50 smaller than that for pre-training, as not to perturb the weights too far from learned solution during pre-training - see Appendix B.3 for further details.

## 3 Results

### 3.1 Ensemble Trajectories

It is not possible to accurately predict a single trajectory due to the stochastic nature of the system, but it is possible to generate an ensemble of equally-likely trajectories. With the ST trained on a range of values of $\beta$, we compare model-generated trajectories with numerical simulations for various parameter values. To reduce training, the ST is only fine-tuned on a limited selection of values of $\beta$ outlined above. We found this was sufficient as the ST is capable of reproducing the expected dynamics on unseen parameter values when interpolating between seen values of $\beta$.

Figure 1 displays the zonally-averaged flow, for four values of $\beta$ unseen during training, with each regime exhibiting distinct dynamics. The ST emulations closely align with reference numerical solutions, with accurate representation of jet nucleation and coalescence frequencies, as well as the latitudinal translation rate. Importantly we observe that the ST does not become physically inconsistent after a number of autoregressive steps.

The comparison between ST emulations and reference trajectories not only confirms the model's adeptness in capturing the system's dynamics but also highlights its robustness across a range of variable parameter values. To provide quantitative assessment of the performance of the ST, we analyse the ability of the ST to reproduce key statistical properties of the stochastically driven jet system. These properties include spatial and temporal correlations, as well as spectral composition, with results shown in Appendix C. The ST's predictions align well with the expected statistical behaviour, providing confidence in its reliability for various scientific and engineering applications.

### 3.2 Uncertainty Quantification: Spontaneous Transition Events

In chaotic or stochastically driven systems predicting transition events, such as nucleation and coalescence events, are of interest, however large ensembles are required for accurate estimate of transition probabilities. As outlined in Shokar et al. (2023), the ST provides a five-order-of-magnitude speedup over traditional numerical methods, facilitating the generation of such ensembles.

Figure 2.a displays a reference coalescence event, while Figure 2.d displays a reference nucleation event, both generated via numerical integration with $\beta = 1.35$, a parameter value not seen during training. Each plot shows dashed lines prior to each event to display the initial conditions used to generate ensembles and measure the time interval between each initial condition and event of interest. In Figures 2.b,c,e,f the ST generates PDFs that show good agreement with the PDFs produced using numerical integration. This demonstrates the ST's capability of effectively capturing the dynamics of the physical system while generalising to unseen regimes. This allows for computationally efficient exploration of physical systems where previously prohibitively expensive.

Sampling the network in this manner provides an estimate of aleatoric uncertainty, with the model naturally demonstrating reduced uncertainty as the forecast approaches the event occurrence. Moreover, it exhibits lower uncertainty when predicting coalescence events compared to nucleation events, consistent with observations from numerical studies in Cope (2021).

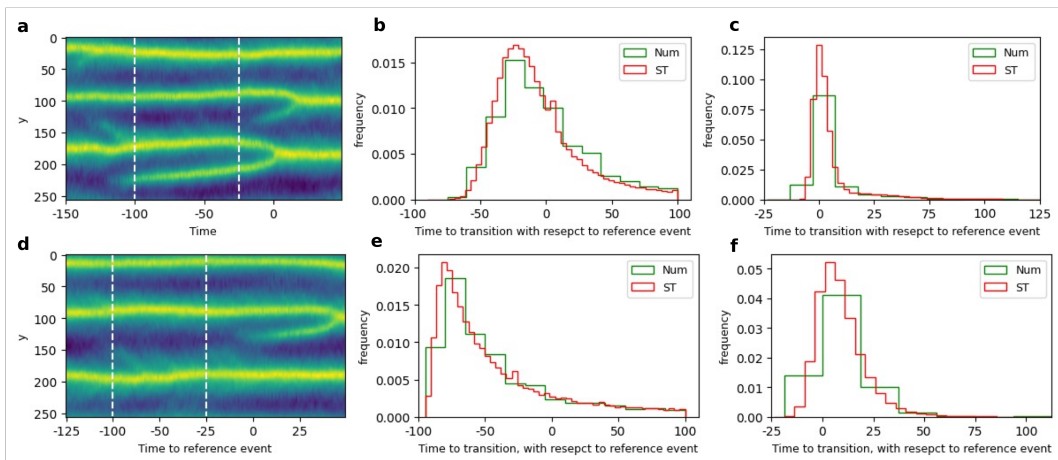

Figure 2: **a**, Reference coalescence event, obtained via numerical integration with $\beta = 1.35$, a parameter value not seen by the ST during training. The two dotted lines indicate initial conditions used to initialise ensembles via numerical integration and the ST. **b**, Probability density functions (PDFs) showing the the time to a coalescence event, with respect to the reference event in shown **a**, with the initial conditions 100 time units before the reference coalescence event. The green curve shows the PDF from numerical integration, comprising an ensemble of size 1000. The red curve shows the PDF from the ST, comprising an ensemble of 100,000. **c**, Shows the same information as **b**, but for initial conditions 25 time units before the reference coalescence event. **d**-**f**, Show the same information as **a**-**c**, but for a reference nucleation event. We observe agreement between the green and red curves indicating that the SLT is successfully capturing the expected distributions.

## 4 CONCLUSION

The fine-tuned Stochastic Transformer is capable of emulating a stochastically driven fluid system, and is successfully shown to generalise to unseen parameter conditions, accurately capturing system dynamics across a spectrum of diverse parameter regimes. Demonstrating its ability to reproduce statistical properties of the system along side its computational efficiency when compared to traditional numerical integration, the ST facilitates economical exploration of the system parameter space, thereby enabling in-depth analyses of spontaneous transition events and other dynamic phenomena.

Future work will look to extending the ST to include varying multiple parameters physical $\{\beta, \mu, \nu, k_f, \varepsilon\}$ to comprehensively characterise the state space of the system. This computationally efficient emulator enables cost-effective exploration of system dynamics, facilitating investigations previously impractical with numerical integration. Tasks include identifying the critical parameter $\beta$ governing transitions between preferred jet numbers and examining how $\beta$ affects the frequency of spontaneous transition event frequencies. Work will also look to expand beyond beta plane turbulence to emulate a diverse array of systems, both deterministic and stochastic, in order to thoroughly evaluate model robustness. Additionally, our aim is to extend the architecture to accommodate 2D and 3D inputs, enabling us to effectively model a broader spectrum of systems.

### ACKNOWLEDGEMENTS

I.S. acknowledges funding by the UK Engineering and Physical Sciences Research Council (grant number EP/S022961/1) as part of the UKRI Centre for Doctoral Training in Application of Artificial Intelligence to the Study of Environmental Risks. We would like to thank the reviewers and editors for taking the time and effort to review and for providing comments and feedback that were invaluable for improving the paper.

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

## A    Numerical simulations of Quasi-Geostrophic Turbulence

Direct numerical simulation of vorticity in equation 1 is performed on a 2D doubly-periodic square domain $(x, y) \in [0, 2\pi]^2$, using a pseudo-spectral method using the GeophysicalFlows (Constantinou et al., 2021) package in Julia (Bezanson et al., 2017) - and details are as in Shokar et al. (2023).

To promote the spontaneous emergence of zonal flows, rather than being directly forced, the fluid is stirred using a stochastic vorticity force $\xi(x, y, t)$ with zero mean injected onto an annulus of wavevectors in Fourier space centred around a mean radial wavenumber $k_f$ with thickness $\delta k = 1$.

Simulations are initialised from rest, $\zeta = 0$, and parameters remain constant throughout the integration. The time for total kinetic energy to reach a statistically steady state during spin-up depends on the damping rate $\mu$, typically achieved by $\mu t = 2 - 3$ (here $t = 50 - 75$ time units). Only data points beyond this point are used for training and testing the deep learning model.

In all experiments values $\mu = 4 \times 10^{-2}$, $\nu = 1$, and an energy injection rate of $\varepsilon = 10^{-4}$, with a principal forcing wavenumber of $k_f = 16$ and a time step of $\delta t = 4 \times 10^{-2}$ were chosen to be in the regimes in which coherent zonal jets form, with the jets exhibiting behaviour with a wide range of time variability.

## B    Stochastic Transformer

### B.1    Architecture

The neural network architecture we employ is the Stochastic Transformer (ST) (Shokar et al., 2023). Figure 3 illustrates the ST architecture that comprises a multi-head (MHA) attention blocks and multi-layer perceptrons (MLP) - that consists of two linear layers and a GELU non-linearity between the two. The attention blocks allows the model to weight the importance of different parts of the input sequence and the conditioning inputs, here $\epsilon$ and $\beta$, when making predictions, while the MLPs allow it to learn complex non-linear relationships. The MHA blocks and MLPs each have skip connections around them to enhance gradient propagation towards the input during the back-propagation phase of training. The details of the how the stochastic and parametric conditioning are applied to the MHA are visualised in Figure 4.

To incorporate temporal information of the sequence of time histories, as is standard with Transformer architectures, a learnable temporal encoding vector (Kazemi et al., 2019) is added to the transformer inputs $U_{t-N:t}$, before the concatenation of $\epsilon$ and $\beta$ to indicate they are not part of the temporal sequence.

Assuming spatial homogeneity for the forcing term $\xi$, the beta-plane system demonstrates latitudinal symmetry, denoted by $\Delta U(y, t) = U(\Delta y, t)$, where $\Delta$ signifies a shift operator in the $y$ dimension. This is result of the system's periodic boundary conditions and forcing symmetries. The transformer itself is not a translation invariant architecture. One approach is to replace each linear layer with a convolution operation. However, to avoid limiting the model's ability to only attend to local spatial connections, we chose to instead use the the method of slices (Budanur et al., 2015a;b) to remove the phase information of the data. This allows the model to learn weights in a phase-invariant manner, independent of any shift $\Delta y$ in the physical space, before reintroducing the phase information.

The method of slices involves first taking a discrete Fourier transform, $\mathcal{F}$, to transform the input data, $U_t$ into the spectral domain, $\hat{U}_t = \mathcal{F}(U_t) \in \mathbb{C}^D$, where $D = N_y$. We obtain the phase information, $\phi$, from the first Fourier mode using $\phi = \arg\left(\hat{U}_t^{(1)}\right)$. The phase is extracted from $U_t$ only, $\phi(t)$, as also removing the phase from the previous time steps in the time history would prevent the transformer from also learning the latitudinal drifting dynamics that we observe in the system. We store this phase and construct a phase-aligned solution, by shifting each of the preceding timesteps with respect to $\phi(t)$, $U'_{t-N:t} = Z_{t-N:t}e^{-\phi(t)\cdot K}$, such that the first Fourier mode of $U'_t$ is a pure cosine and $K = [-N_y/2 - 1, ..., N_y/2] \in \mathbb{Z}^D$ is the wavenumbers in spectral space. $U'_{t-N:t}$ is then input to the ST to obtain a prediction of $U'_{t+1}$, this way the ST handles inputs that have been translated to be identical manner, given any shift $\Delta y$. We reintroduce the phase information to the output from the transformer $U'_{t+1}$ via $\phi(t)$, $U_{t+1} = U'_{t+1}e^{\phi(t)\cdot K}$, before an inverse Fourier

**Stochastic Transformer (ST)**

Figure 3: Schematic of the Stochastic Transformer (ST) architecture that incorporate $\beta$ and $\epsilon \sim \mathcal{N}(0,1)$. Arrows indicate the forward pass, where the ST receives an input including the value of $\beta$ and a short temporal history of $U_{t-N,t}$, with $N$ denoting the sequence length. The model's weights are learned independently of the phase, $\phi$, which represents the argument of the first mode of $\hat{U}_t \in \mathbb{C}^D$. Noise vector, $\epsilon$ is appended to the space-time histories. The architecture comprises M transformer blocks, where 'MHA' stands for multi-headed attention using scaled dot-product attention - see Figure 4 for details. 'Norm' refers to layer normalisation, and 'MLP' is a position-wise multi-layer perceptron. The phase information is reintroduced to produce a forward trajectory of $\tilde{U}_{t+1}$.

transform, to transform the outputs back to physical space, $U_{t+1} = \mathcal{F}^{-1}(U'_{t+1}) \in \mathbb{R}^D$. This is visualised in Figure 3. As such the model acts in a translation equivalent manner. We found that including this symmetry was important for producing the correct PDFs shown in Figure 2.

### B.2 OBJECTIVE FUNCTION

The ST is pre-trained and fine-tuned using the following loss function, comprising of the Continuous Ranked Probability Score (CRPS) (Matheson & Winkler, 1976), a proper scoring rule that generalises the Mean Absolute Error (MAE), and a spectral loss to ensure that energy is preserved at all scales:

$$\mathcal{L} = \text{CRPS}\left(U_{t+1}, \tilde{U}_{t+1}\right) + \lambda \, \text{MAE}\left(|\mathcal{F}[U_t]|, |\mathcal{F}[\tilde{U}_t]|\right) \tag{7}$$

The $\text{CRPS}(U, \tilde{U}) = \frac{1}{m}\sum_{i=1}^{m}\left|U - \tilde{U}^{(i)}\right| - \frac{1}{2m^2}\sum_{i=1}^{m}\sum_{j=1}^{m}\left|\tilde{U}^{(i)} - \tilde{U}^{(j)}\right|$ is commonly used metric for probabilistic forecasts in meteorology and was demonstrated by (Pacchiardi et al., 2022) to an effective alternative to adversarial training methods. Here $U$ is the truth trajectory from the training dataset, $\tilde{U}^{(i)}$ is the $i^{th}$ member of the prediction ensemble, where the model produces an ensemble of size $m$ for time step $t + 1$. The network is trained to minimise the difference between each predicted ensemble member and the truth trajectory $U_{t+1}$, while simultaneously maximising the dissimilarity between each individual ensemble member $\tilde{U}_{t+1}^{(i)}$ given to the inclusion of the second term in the CRPS. While performing a forward pass over an an ensemble increases training costs, in practice an $m = 2$ is sufficient.

Neural networks have been shown to exhibit a spectral bias (Rahaman et al., 2019), inadequately capturing higher frequency modes due to their diminished contribution to overall energy compared to the larger-scales. In the context of modelling physical data, the preservation of not only large-scale properties but also energy across all scales is essential. For this reason a spectral loss is employed, this involves employing the discrete Fourier transform, denoted as $\mathcal{F}$, to transform $U_t$ and $\hat{U}_t$ and subsequently taking the modulus, resulting in $|\mathcal{F}[U_t]| \in \mathbb{R}$ and $|\mathcal{F}[\tilde{U}_t]| \in \mathbb{R}$, before taking the MAE

Figure 4: The multi-headed attention block. The initial transformer block among the M blocks incorporates a stochastic and parametric variant of multi-headed attention - the layer is similar to the standard attention block, with the addition of forcing noise $\epsilon \sim \mathcal{N}(0, 1)$, and conditioning parameter $\beta$. $\beta$ is shaped in order to be concatenated with the input time histories, $U_{t-N:t}$ across the feature dimension, for the $K$ (Key), and $V$ (Value) vectors, while $N$ represents the length of the latent time history. Weights $W^K$ & $W^V$ independently linearly transform these, where $D$ denotes the feature dimension. In the stochastic variant, additional weights $W^{K_\epsilon}$ & $W^{V_\epsilon}$ linearly transform $\epsilon$, before concatenation to give the $K$ and $V$ vectors. The query (Q) vector is simply a linear transformation ($W^Q$) of the input time histories $U_{t-N:t}$. Dashed lines indicate the additional steps due in the stochastic variant, dotted lines indicate the additional steps due in the parametric variant. $A$ is the attention matrix - a mask is applied to the upper triangular elements to prevent forward looking, as is applied to the decoder transformer in Vaswani et al. (2017), before being scaled by $\sqrt{D}$ and a softmax transformation is applied. This illustrates the case where the batch size = 1 and the number of heads = 1. The non-stochastic parametric self-attention in the following transformer blocks operates in the exact same manner, simply without the transformation and concatenation of $\epsilon$, but still include concatenation with $\beta$.

between these two quantifies. Through experimentation, we found $\lambda = 1$ to weight these terms achieved convergence during training.

While a time step of $\delta t = 4 \times 10^{-4}$ is required for stability of the numerical integration, sampling at such small intervals is unnecessary for the neural network to capture the dynamics. Therefore data input to the network is sampled at time intervals 2500 times larger than the numerical integration time step. Each time step of the system is represented as a one-dimensional vector $U_t(t) \in \mathbb{R}^D$, with the target output being the subsequent time step $U_{t+1}(y)$, also obtained from the numerical integration.

### B.3 MODEL HYPERPARAMETERS

Using a Bayesian hyperparameter sweep (Biewald, 2020) we determine the best-performing model. The metric used to minimise was the compose loss function outlined in equation 7. Through this sweep we found that the following hyperparameters were optimal: batch size: 256; optimiser: Adam; initial pre-training transformer learning rate: 5e-4; initial fine-tuning transformer learning

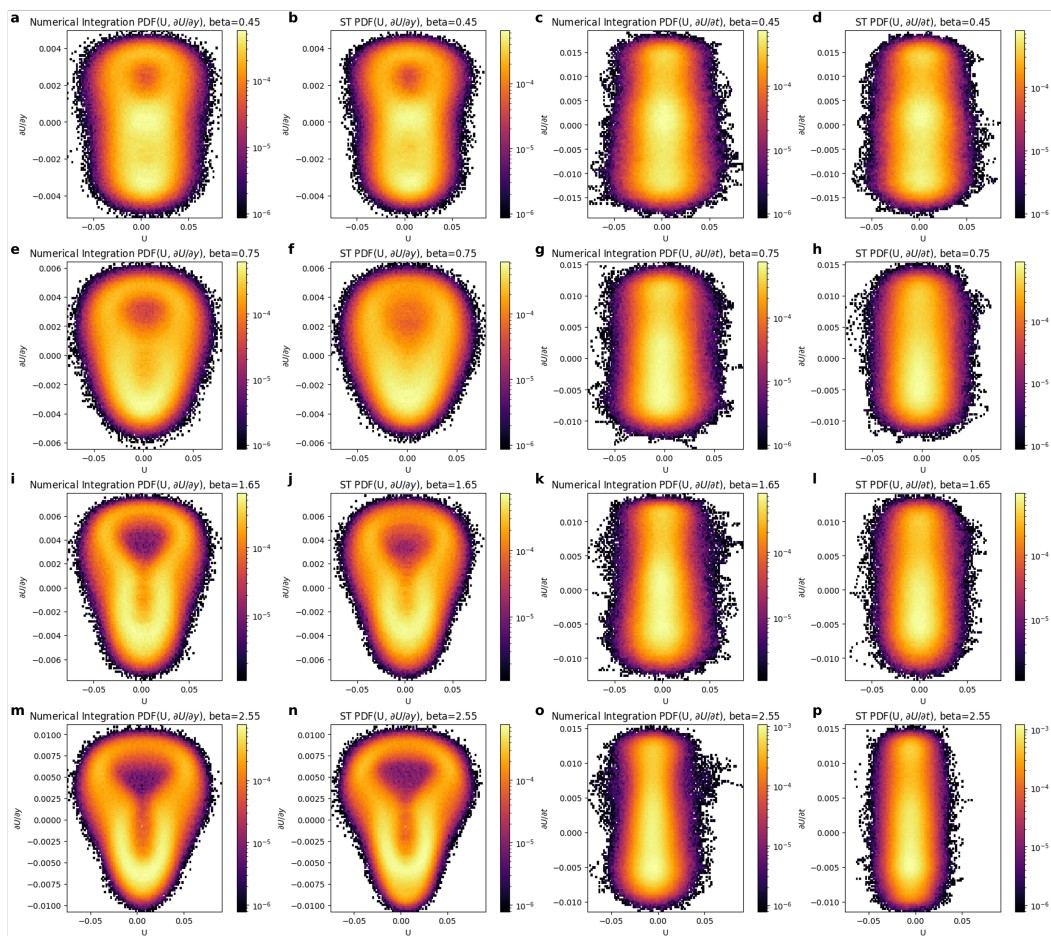

Figure 5: PDFs displaying joint probabilities of $U$, $\partial_y U$ and $\partial_t U$. **a**, **e**, **i**, **m**, for visualisation of the 3D PDF we average over $\partial_t U$ to obtain the density of values in the $u$-$\partial_y U$ space obtained via numerical integration for different values of $\beta = \{0.45, 0.75, 1.65, 2.55\}$ unseen during training. **b**, **f**, **j**, **n** show the same $u$-$\partial_y U$ space from an evolution generated using the ST. **c**, **g**, **k**, **o** show the $u$-$\partial_t U$ space by averaging over $\partial_y U$ for the numerical integration values and **d**, **h**, **l**, **p** show the $u$-$\partial_t U$ space for evolution generated using the ST. Here we can see that again the SLT shows good agreement with the numerical integration, covering the full space of dynamics, across the parameter range.

rate: 1e-5, learning rate scheduler: exponential decay with decay rate: $\gamma$=0.99; pre-training epochs: 250, fine-tuning epochs: 250, CRPS ensemble size: 2; $N$ (length of time history used to make next forecast): 2, number of transformer blocks: 4, number of transformer heads: 16.

## C  VALIDATION METRICS

For a quantitative assessment of longer-term evolutions, we examine statistical properties of the flow. We generate Probability Density Functions (PDFs) for individual $U$ values and their derivatives, $\partial_y U$ and $\partial_t U$. These PDFs serve to measure the learned spatial and temporal correlation. We conduct this analysis over a time integration period of 5000 time units, producing two PDFs: $p(U)$ representing the PDF from numerical integration, $U$, and $q(\tilde{U})$ representing the PDF of forecasted values derived from the SLT, $\tilde{U}$.

In Figures 5 we plot the joint PDFs of $U$, $\partial_y U$ and $\partial_t U$. For visualisation of the 3D PDFs, we sum over $\partial_t U$ to obtain the density of values in the $u$-$\partial_y U$ space, shown in Figure 5.a,e,i obtained

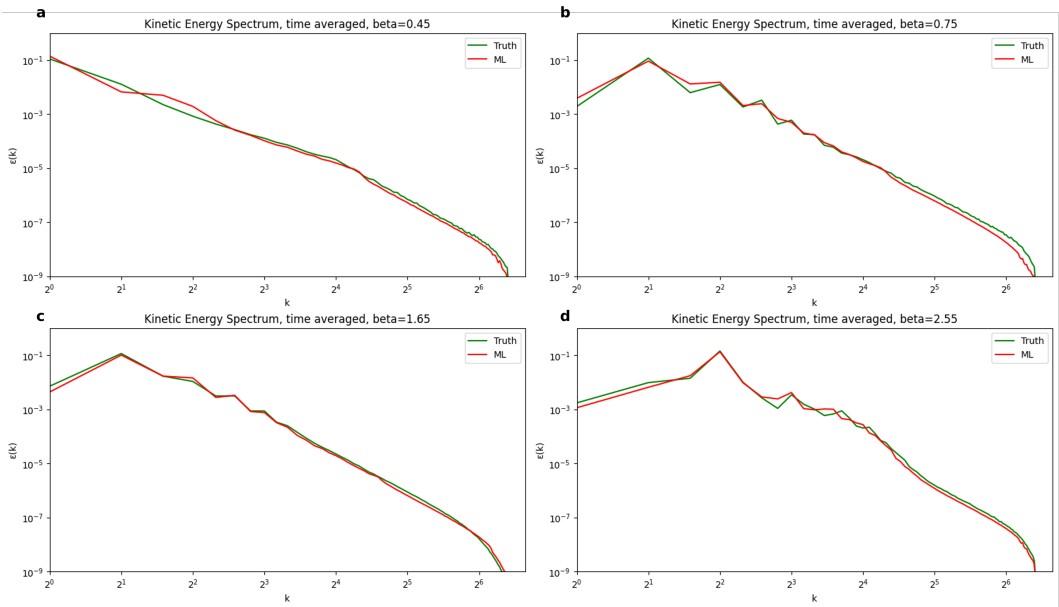

Figure 6: Time averaged power spectral density displayed for both the numerical integration (green) and the ST (red), showing the energy content of each wavenumber in the zonally-averaged zonal direction, $U$, for values of $\beta = \{0.45, 0.75, 1.65, 2.55\}$, averaged over 5,000 time units. Again the ST shows good agreement with the numerical integration for values of $\beta$ unseen during training.

via numerical integration and Figure 5.b,f,j obtained from the ST. Similarly, we show the density of values in the $u$-$\partial_t U$ space, by summation over $\partial_u U$, for numerical integration in Figure 5.c,g,k and the ST in Figure 5.d,h,l. We see good agreement between the outputs from the ST and numerical integration, across the different parameter values, covering the full space of dynamics.

As $\beta$ increases, we do observe discrepancy between the PDFs generated via numerical integration and the ST for the joint distribution of $U - \partial_y U$. We hypothesise that this is due to a reduced frequency of spontaneous events as zonostrophy increases, leading to a data imbalance. Future work shall explore this phenomena, however overall we see very good agreement between the numerical integrations and the ST emulations.

The proportion of energy in each wavenumber, or the energy spectra, is an important physical property in fluid dynamics. For this we assess time-averaged power spectral density plotted for both the numerical integrations (in green) and the the ST (in red) over the same time integration period of 5000 time units as used above, plotted in Figure 6. We observe a high level of agreement across all scales, paying particular attention on the higher wavenumbers where data-driven approaches often fail. This is all the more important in the case of 2D turbulence due to an inverse energy cascade. This means that correct representation of the small scales is vital due to their upstream influence on the large scale dynamics.

These findings, in conjunction with the aforementioned evaluation metrics, allow one to place confidence in the ST's capacity to faithfully emulate the system dynamics, over the range of parameter values, even for those unseen during training. We believe this positions the ST as a valuable tool for investigating dynamical phenomena observed in the beta-plane system, facilitated by its computational speed-up.

