# OpenReview forum: "Extending Deep Learning Emulation Across Parameter Regimes to Assess Stochastically Driven Spontaneous Transition Events"
_ICLR.cc/2024/Workshop/AI4DiffEqtnsInSci — AI4DiffEqtnsInSci @ ICLR 2024 Poster_

### Official Review · Reviewer_jZDE · 2024-02-25
**Review of Extending Deep Learning Emulation Across Parameter Regimes to Assess Stochastically Driven Spontaneous Transition Events**

**Rating:** 8
**Confidence:** 4

**Review:**

The paper discusses the extension of deep learning emulation for assessing stochastic events across different parameter regimes, focusing on spontaneous transition events driven by stochastic factors. It employs transformer-based networks to emulate complex dynamical systems, highlighting a significant computational cost reduction and the ability to generalize across diverse scenarios. This research presents a novel approach in combining deep learning with stochastic dynamics, aiming to enhance the understanding and prediction of unpredictable events in various scientific domains.

The manuscript presents an innovative integration of deep learning with stochastic dynamics, using the Stochastic Transformer (ST) to emulate complex systems across various parameter regimes. This approach not only demonstrates a significant computational efficiency but also extends the model's generalization capabilities to novel conditions. The use of pretraining for the SST across a broad spectrum of parameters showcases an interesting application of current PDE research with deep learning in stochastic modeling.

However, the application to a single system, though understandable given scope constraints, leaves open the question of the methodology's scalability and effectiveness across diverse domains. Future work could benefit from exploring a range of systems to validate the approach's universality, as also highlighted in the final section of the paper.

Minor comment:
- Second sentence in 4. Conclusions: typo for "it's" where it should be "its".

Overall, the paper is articulate and detailed, effectively communicating its contributions and findings. Given its novel application and the depth of analysis, it would undoubtedly spur valuable discussions at the workshop. I recommend acceptance of this work.

---

### Official Review · Reviewer_i9ZV · 2024-02-28
**Octavi's review**

**Rating:** 4
**Confidence:** 4

**Review:**

I don't see the novelty of the paper. The authors pre-train a transformer and then fine-tune for task specific downstream problems.
The novelty might be in the use case - stochastic dynamical systems - but there is a lot of good work in this topic that is not even mentioned here. Also, the writing could be improved, it is hard to understand.

For instance:
Alexandre Cortiella, Kwang-Chun Park, Alireza Doostan,
Sparse identification of nonlinear dynamical systems via reweighted ℓ1-regularized least squares
Computer Methods in Applied Mechanics and Engineering

---

### Meta-Review · Area_Chair_rTJc · 2024-03-01

**Recommendation:** Accept (Poster)

**Metareview:**

Authors proposes using a pre-trained transformer model fine-tuned on task-specific data to emulate complex stochastic dynamical systems. Paper shows improved computational efficiency and generalization capabilities. I recommend authors provide details on model training and improving the clarity of the paper for the camera-ready version. If possible, I encourage authors to expand the experiments beyond a single system and compare to prior work in the area.

---

### Decision · Program_Chairs · 2024-03-02

Accept (Poster)